# Advanced load frequency control of microgrid using a bat algorithm supported by a balloon effect identifier in the presence of photovoltaic power source

**Ahmed M. Ewias**[1], **Sultan H. Hakmi**[2], **Tarek Hassan Mohamed**[1], **Mohamed Metwally Mahmoud**[1]*, **Ahmad Eid**[3,4], **Almoataz Y. Abdelaziz**[5], **Yasser Ahmed Dahab**[6]

1 Department of Electrical Engineering, Faculty of Energy Engineering, Aswan University, Aswan, Egypt,
2 Electrical Engineering Department, Faculty of Engineering, Jazan University, Jazan, Saudi Arabia,
3 Department of Electrical Engineering, Faculty of Engineering, Aswan University, Aswan, Egypt,
4 Department of Electrical Engineering, College of Engineering, Qassim University, Unaizah, Saudi Arabia,
5 Faculty of Engineering and Technology, Future University in Egypt, Cairo, Egypt, 6 Arab Academy for Science, Technology and Maritime Transport, South Valley Branch, Aswan, Egypt

* Metwally_M@aswu.edu.eg

**Data Availability Statement:** All relevant data are within the paper.

## Abstract

Due to the unpredictability of the majority of green energy sources (GESs), particularly in microgrids (µGs), frequency deviations are unavoidable. These factors include solar irradiance, wind disturbances, and parametric uncertainty, all of which have a substantial impact on the system's frequency. An adaptive load frequency control (LFC) method for power systems is suggested in this paper to mitigate the aforementioned issues. For engineering challenges, soft computing methods like the bat algorithm (BA), where it proves its effectiveness in different applications, consistently produce positive outcomes, so it is used to address the LFC issue. For online gain tuning, an integral controller using an artificial BA is utilized, and this control method is supported by a modification known as the balloon effect (BE) identifier. Stability and robustness of analysis of the suggested BA+BE scheme is investigated. The system with the proposed adaptive frequency controller is evaluated in the case of step/ random load demand. In addition, high penetrations of photovoltaic (PV) sources are considered. The standard integral controller and Jaya+BE, two more optimization techniques, have been compared with the suggested BA+BE strategy. According to the results of the MATLAB simulation, the suggested technique (BA+BE) has a significant advantage over other techniques in terms of maintaining frequency stability in the presence of step/random disturbances and PV source. The suggested method successfully keeps the frequency steady over I and Jaya+BE by 61.5% and 31.25%, respectively. In order to validate the MATLAB simulation results, real-time simulation tests are given utilizing a PC and a QUARC pid_e data acquisition card.

**Funding:** The author(s) received no specific funding for this work.

**Competing interests:** The authors have declared that no competing interests exist.

**Abbreviations:** RESs, Renewable energy sources; ESTs, Energy storage technologies; LFC, Load frequency control; BA, Bat optimization algorithm; PI-FOPID, PI-fractional order PID; FD, deviations in frequencies; VSG, virtual synchronous generator; MPC, Model predictive control; RT, Real-time; μG, Microgrid; PV, Solar; BE, Balloon effect; PS, power system; ECs, electric cars; P, Active power; Q, Reactive power; I, integral.

## 1. Introduction

### a) Motivation and background

The primary causes of the current climate are conventional power-generating methods, especially those that rely on fossil fuels [1, 2]. It is of considerable interest to scientists and academics to concentrate on energy production and conservation utilizing clean and renewable energy sources (RESs), such as solar (PV) and wind [3–5]. Because RESs are affordable and simple to use, there has been a lot of interest in them. The power system (PS) is seeing more "uncertainty" as RESs are growing. A viable scheme for integrating disparate RESs, ccontrollable energy storage technologies (ESTs), and load is the microgrid (μG). In contrast, it is more vulnerable to power mismatch than the traditional PS, especially in isolated μG situations. Because to both the intermittent nature of RES and the low inertia of inverter-interfaced RESss, isolated μG's frequency is prone to vary from the permitted range without the help of the PS, which poses significant issues for maintaining its frequency [6, 7].

Primary frequency regulation (FR) such as the droop and virtual synchronous generator (VSG) control, where they offering simple implementation and minimal communication needs, and secondary FR have been typically used to maintain frequencies [8, 9]. Nevertheless, using just primary FR, it is impossible to prevent deviations in frequencies (FD) from the rating value. The FD, for which PID controller-based technique is frequently used, can be eliminated by secondary FR [8, 10]. To reduce the steady-state error, additional control loops are also introduced to the PFR method. A VSG structure was presented in which a virtual regulator works together with the resilient regulator to eliminate the disruption in the secondary FR [11]. This structure was enhanced by the notions of the virtual rotor and virtual the primary & secondary FR. In [12], quantitative feedback approach was used for tuning the VSG's parameters, allowing the load frequency control (LFC) problem to be solved even if the remote μG inertia was significantly reduced.

### b) Literature review

The usage of RESs is anticipated to rise while thermal energy, which formerly dominated, is predicted to become less prevalent. Additionally, it is crucial to investigate a μG-FR because of its high absorption capacity and possible influence. Therefore, it is critical to utilize a controller in an isolated μG that is reliable and effective under a range of circumstances. According to realistically acceptable dynamic performance, LFC keeps the system's power balance within predetermined parameters where it deviates from its nominal value [13, 14]. There are two operating modes for the μG: on-grid and off-grid. ESTs, energy sources, and artificial virtual inertia are a few of the strategies frequently employed in off-grid mode to maintain frequency [15]. One μG's capacity is restricted, and it can be affected by a number of different nonlinear random fluctuations, where a two area mixed μG with solar thermal generator, biodiesel generators, storage elements, and a DC bus where examined for its ability to adjust frequency [16, 17].

Instability and complexity are caused by the use of RESs in PSs. The key μG input elements in PSs are the growth of economic and environmental challenges as well as the dependability of conventional PSs [18]. With integrated controller I, which is frequently utilized in LFC applications, an LFC device's gain may be changed offline. When loading changes and systems are modified, the system performs poorly dynamically. This issue has been resolved using PI controllers with fixed settings [19, 20]. The majority of the "μGs" are made up of diesel generators, RESs, ESTs, and other equipment, and there are power connections between them that may significantly increase security. However, there are additional challenges in synthesizing the system, controlling energy sources, and creating and controlling the structure since its

topology is more complex than the normal μG [21, 22]. Naturally, RESs cause frequency variations and voltage changes in the distribution system. Furthermore, improper management of these sources has negative effects on electricity systems. Because of these factors, it should be necessary to find efficient methods to keep the features of frequency fluctuations and voltage changes within a given range [23]. Based on the details in [24], where models have been presented based on the model design and communication system, a comparison of the models' costs, dependability, and consistency has been made.

However, without adjustments, those systems are unable to cope with the natural unpredictability of the actual μG, and typical functioning limits are all but ignored. Model predictive control (MPC) was demonstrated effectiveness as a method of control strategy for mitigating the drawbacks imposed on by the μG supply/demand instabilities along with system limits, like generation ramping rates and ability [25, 26]. An adaptive fuzzy MPC for secondary FR was proposed in [25], enabling electricity consuming balanced to be ensured despite incoming power and load uncertainty. An MPC approach was given in [27], to accomplish secondary FR in a μG linked with multiple sources and connected electric cars (ECs), where the impact of changing system settings is reduced. For networked μGs, a structure integrating the decentralised MPC & the voltage-based FR was suggested in [28], which realizes FR while fulfilling the bus voltage limitation. According to the most recent studies, MPC's key characteristics help with FR. A thorough control structure is suggested in [29] that guarantees the effectiveness of FR while greatly reducing the added harmonics of converters. Ref. [30] introduces the idea of the highest possible loading factor and afterwards suggests a dispersed secondary FR technique with low-bandwidth signaling for μG in order to attain FR and precise active power (P) pooling. Ref. [31], it presents a compromise theory for voltage control and reactive power (Q) exchange, and proposes a decentralised controller for secondary FR and voltage management in μG. Ref. [32] developed a multipurpose completely distributed command architecture that enables voltage/FR and P/Q exchange in inverter-based μGs.

Many industrial applications have employed the bat optimization algorithm (BA) to adaptively adjust the gains of traditional controllers [33]. A cascaded PI-fractional order PID (PI-FOPID) controller with fine-tuned BA can improve the hybrid μG system frequency response [34]. In order to solve this issue and increase the optimization algorithm's sensitivity to both disturbances and parameter changes, a balloon effect (BE) adjustment proved its effectiveness as provided in [35] so, it is suggested for this study.

## c) Problem formulation

The adaptive control problem is shown using several optimization strategies. These techniques may also be used to fine-tune the parameters of fuzzy controllers or neural network controllers, as shown in [36–38], and in these cases, the object function (OF) is determined by the error value of the controlled variable. The settings of the adaptive controller may also be tuned directly using optimization approaches, as indicated in [39, 40]. However, in these attempts, the OF was constructed, for instance, using temporal response characteristics such as rising time, settling time, and overshoot.: $J_{min} = \Sigma (M_P + T_s + T_r)$.

The fact that $M_P$, $T_s$ and $T_r$ are functions in the nominal values of the system parameters, however, is a drawback of this approach, particularly in the case of time-variant systems. A BE change has just been proposed to address this issue [39, 40]. The goal of BE is to have the OF interact with the updated parameter values and other system modifications. In short, basic optimization methods may be used to modify control parameters in a variety of real-world and commercial applications, including motor control, temperature control, etc.

### d) Contributions

This research suggests an innovative adaptive LFC technique to enhance the degree of PV participation in μG in order to address the aforementioned difficulties. This paper proposes an adaptive LFC scheme for fluctuating loads and parameters in smart μG, based on a BA with BE (BA+BE). Diesel generators, electrical load, and PV make up the μG that is being considered. The influence of FD brought on by both random demand loads and RESs is investigated in order to evaluate the proposed (BA+BE) optimizer. In order to demonstrate its accuracy and robustness, it is also contrasted with the traditional integral (I) controller and Jaya approaches. A thorough simulation and real-time (RT) investigation are undertaken to confirm the successful application of the idea, and the suggested control strategy's detailed design process and implementation structure are described. A laboratory implementation of the desired controller with the studied system is presented. In this step, the BA+BE, I, and Jaya +BE algorithms are applied to RT implementation using a PC with QUARC pid_e data acquisition card and MATLAB software with QUARC sub-program. Using a storage oscilloscope, the system frequency and algorithm outputs are recorded. The main outstanding features of this work can be expressed as follows:

- Using the BA+BE optimizer, which is fed by the output of the open-loop simplified μG transfer function, an online adaptable LFC is investigated.

- This paper demonstrates the efficiency of a BA+BE optimizer-adjusted I controller in LFC issues.

- The performance of the suggested adaptive approach is superior to that of the traditional I and Jaya+BE controllers.

- Stability and robustness of analysis of the suggested BA+BE scheme is investigated.

### e) Paper organization

The rest of the paper is organized as follows: Section 2 discusses the applied techniques including the standard BA, and a brief description of BE. Section 3 describes the islanded single area μG dynamic model and presents its configuration. Section 4 presents the stability and robustness of analysis of the investigated controller. Section 5 offers the simulation results and discussions of the system with the proposed controlled systems. In addition, a RT implementation for the studied μG is presented in the same section. Finally, Section 6 concludes the work.

## 2. Applied techniques

### 2.1. BA technique

As eye-catching creatures, bats have piqued the interest of researchers from a variety of fields due to their superior echolocation abilities. A type of sonar called echolocation is used by bats, mostly microbats, to determine the distance to an item by listening for the echoes that return to their ears after they emit a loud and brief pulse of sound [41]. The ability to distinguish between an obstruction and a target thanks to this extraordinary placement technique enables bats to hunt even in complete darkness. Yang, Xin-She, was inspired to create the BA by how bats behave. Yang created the BA, a population-based metaheuristic algorithm, to address challenges involving continuous optimization [42].

 The fundamental BA was developed using biological inspiration from bats' echolocation or bio-sonar traits. For the purpose of hunting or navigation, bats in nature emit ultrasonic waves into their immediate surroundings. After these waves are emitted, it receives their echoes, and

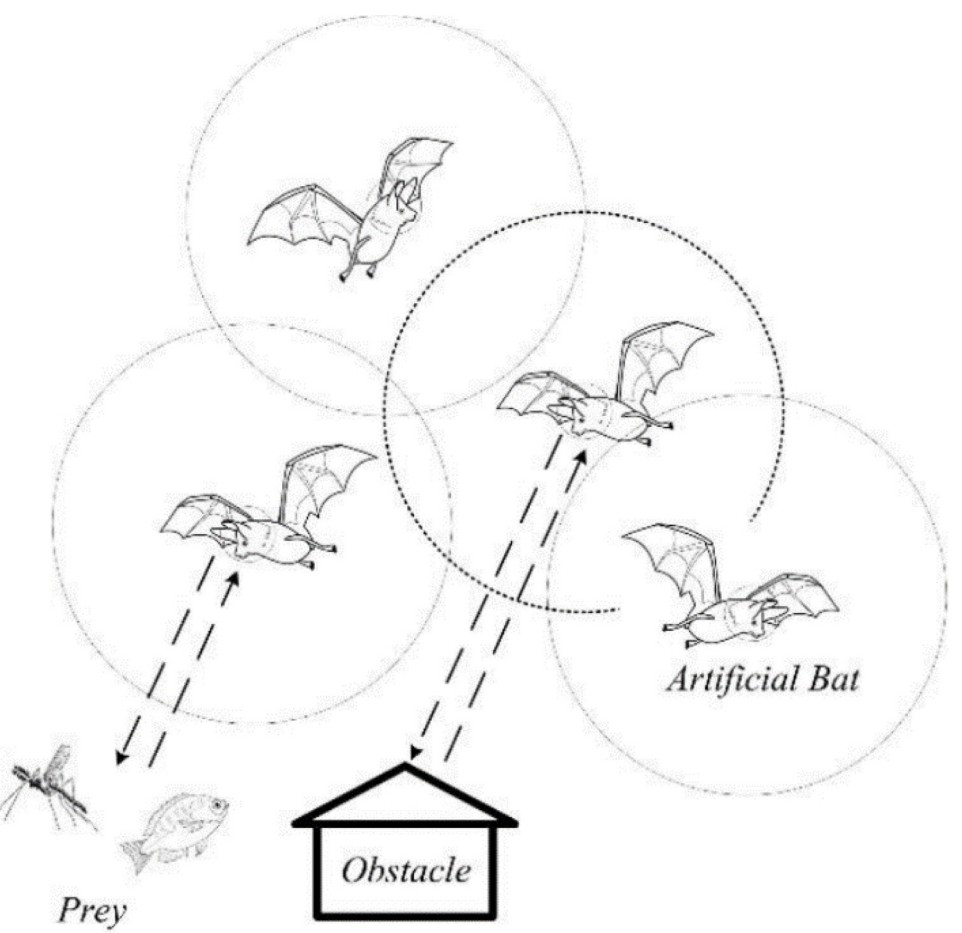

**Fig 1. Bat behavior.**

using the echoes it gets, as illustrated in Fig 1, it uses them to locate itself, detect impediments in its path, and identify prey. Furthermore, each agent in the swarm is capable of traveling to a prior best position that the swarm previously discovered or locating the most "nutritious" locations. The BA has demonstrated exceptional effectiveness in solving continuous optimization issues [43].

The bat population must first be initialized before the pulse frequency can be determined. Next, the pulse rates and loudness must be initialized, and finally, the maximum number of repetitions must be determined. If the outcome is improved, new values will be generated and the values will be updated in velocities. In this case, random values will be generated; if a solution is found, we must choose the best one; otherwise, the software will go back. It receives newly formed solutions in the form of random values, finds the best current value, and produces the result [42]. Fig 2 depicts the BA Flowchart.

The first thing that happens is that every bat has its initial location, velocity, and frequency. The movement of the virtual bats is determined by updating their location and velocity using Eqs 1, 2, and 3 as shown below for each time step t, where T is the maximum number of iterations [42].

$$f_i = f_{min} + (f_{min} - f_{max})\beta \tag{1}$$

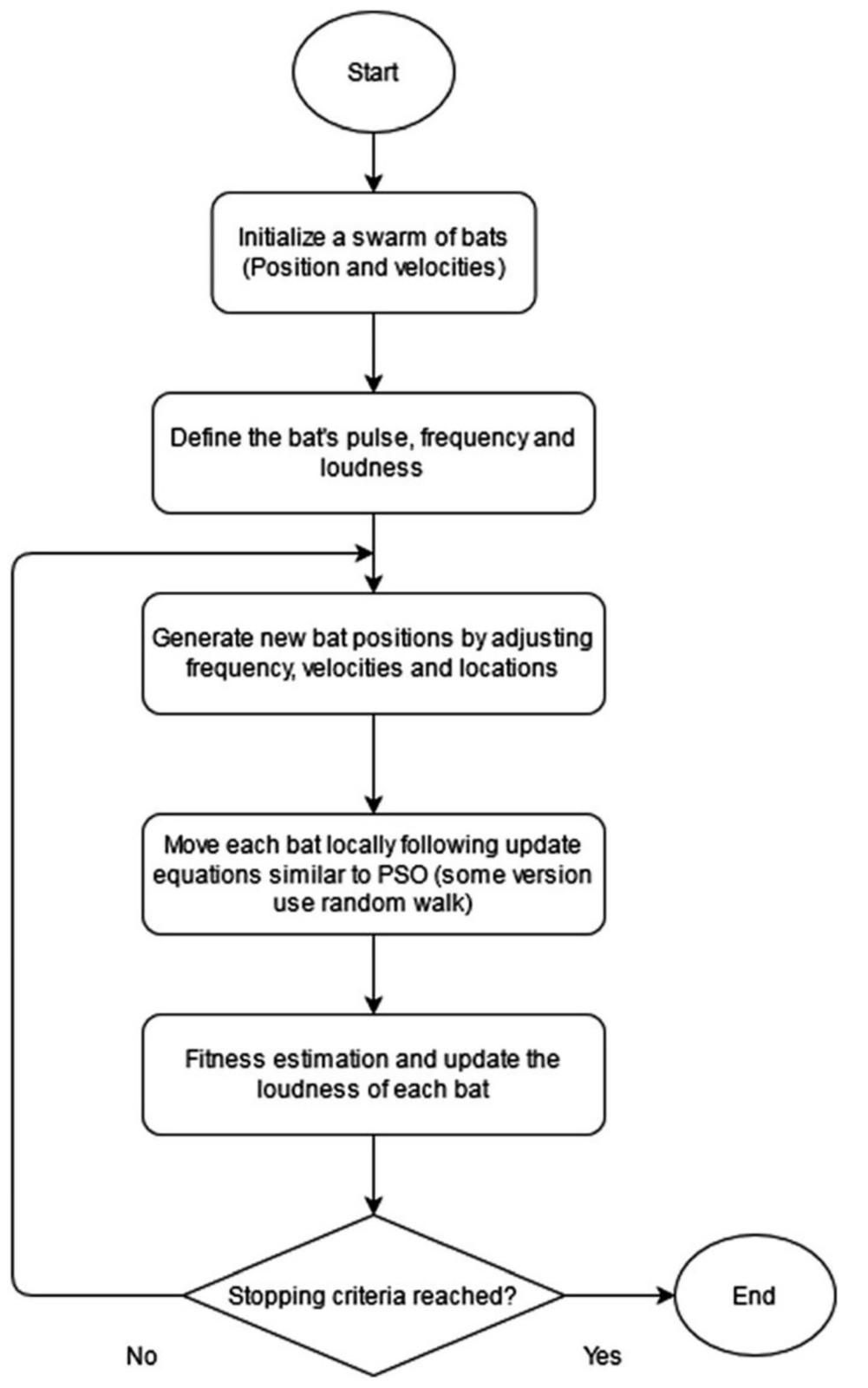

**Fig 2. BA flowchart.**

$$v_i^t = v_i^{t-1} + [x_i^{t-1} - x_*]f_i \tag{2}$$

$$x_i^t = x_i^{t-1} - v_i^t \tag{3}$$

where $\beta \in [0, 1]$ indicates a random vector chosen from a uniform distribution, $f_i$ the frequency of each bat and is the current global best solution (location) $x_*$.

In this case is found by comparing all of the solutions among all n bats at each iteration. After the bats' positions are updated, a random number is formed; if the random number is greater than the pulse emission rate $r_i$, a new location will be generated around the most effective solutions at the time. This new position may be represented by Eq (4) [43].

$$x_{new} = x_{old} + \varepsilon A^t \tag{4}$$

where, $\varepsilon \in [-1,1]$, is a random number, while $A^t$ is the average loudness of all the bats at the current iteration. Furthermore, the loudness $A^t$ and the pulse emission rate $r_i$ will be updated and a solution will be accepted if a random number is less than loudness $A^i$ and $f(x_i) < f(x_*)$. $A^i$ and $r_i$ are updated by (5)

$$A_i^{t+1} = \alpha A_i^t, r_i^{t+1} = r_i^0[1 - e^{-\gamma t}] \tag{5}$$

The procedure iterates until the termination requirements are satisfied

## 2.2. Investigated BE identifier

The BE is defined as how air affects balloon size. System difficulties like disturbances and parameter changes can significantly affect $G_i(s)$, much like the balloon effect did. Fig 3 illustrates how an optimization strategy's objective function is impacted by the BE identifier at each iteration. By employing this strategy, the algorithmic process is improved [35, 44]. The online transfer function of microgrid for any iteration (i) will be:

$$G_i(\text{s}) = \frac{Y_i(s)}{U_i(s)} \tag{6}$$

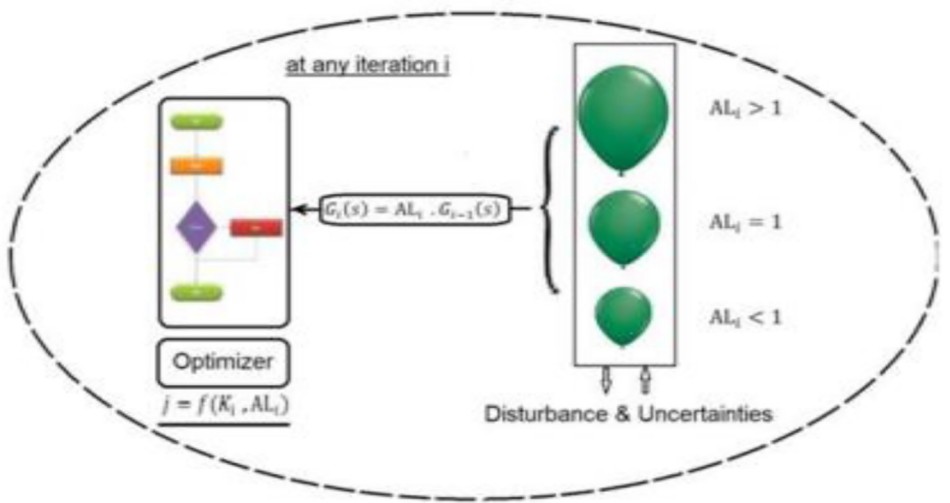

**Fig 3. An optimization strategy-based identifier for the balloon effect [44].**

$G_i(s)$ is also a function of its preceding value $G_{i-1}(s)$. $G_0(s)$ denotes the nominal process transfer function, and $AL_i$ stands for a gain.

$$G_i(s) = AL_i G_{i-1}(s) \tag{7}$$

$$G_{i-1}(s) = \rho_i G_0(s) \tag{8}$$

where

$$\rho_i = \prod_{n=1}^{i-1} AL_n \tag{9}$$

$$G_i(s) = AL_i \rho_i G_0(s) \tag{10}$$

The main advantage of mixing the BE identifier with the optimization method in adaptive case is to absorb large range of system parameters uncertainties and system disturbances, this leads to improve computation process and decrease its burden.

## 3. Power system dynamic model

Fig 4 shows the block diagram of a µG-PS. The dynamic model of the proposed µG-PS can be described in the following equations [41]: The total Load-generator dynamic relationship between the supply error $\Delta P_d$ -$\Delta P_L$ and the frequency deviation ($\Delta f$) can be expressed as:

$$\Delta f = \left(\frac{1}{M}\right).\Delta Pd - \left(\frac{1}{M}\right). - \Delta PL - \left(\frac{D}{M}\right).\Delta f \tag{11}$$

The diesel generator' dynamic can be expressed as:

$$\Delta Pd = \left(\frac{1}{Td}\right).\Delta Pg - \left(\frac{1}{Td}\right).\Delta Pd \tag{12}$$

The governor's dynamic can be expressed as:

$$\Delta Pg = \left(\frac{1}{Tg}\right).\Delta Pc - \left(\frac{1}{R.Td}\right).\Delta f - \left(\frac{1}{Tg}\right).\Delta Pg \tag{13}$$

The symbols appear in Fig 4 is defined as follows: $\Delta P_g$: The governor output change, $\Delta P_d$: The

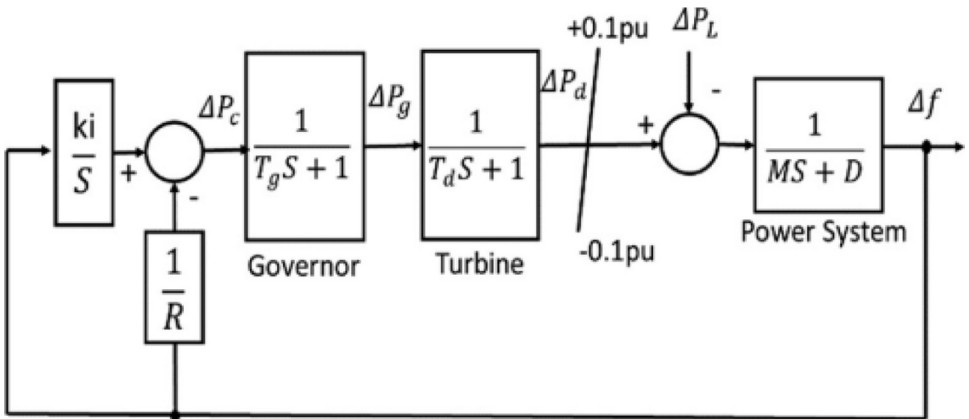

**Fig 4. Block diagram of the model of µG-PS.**

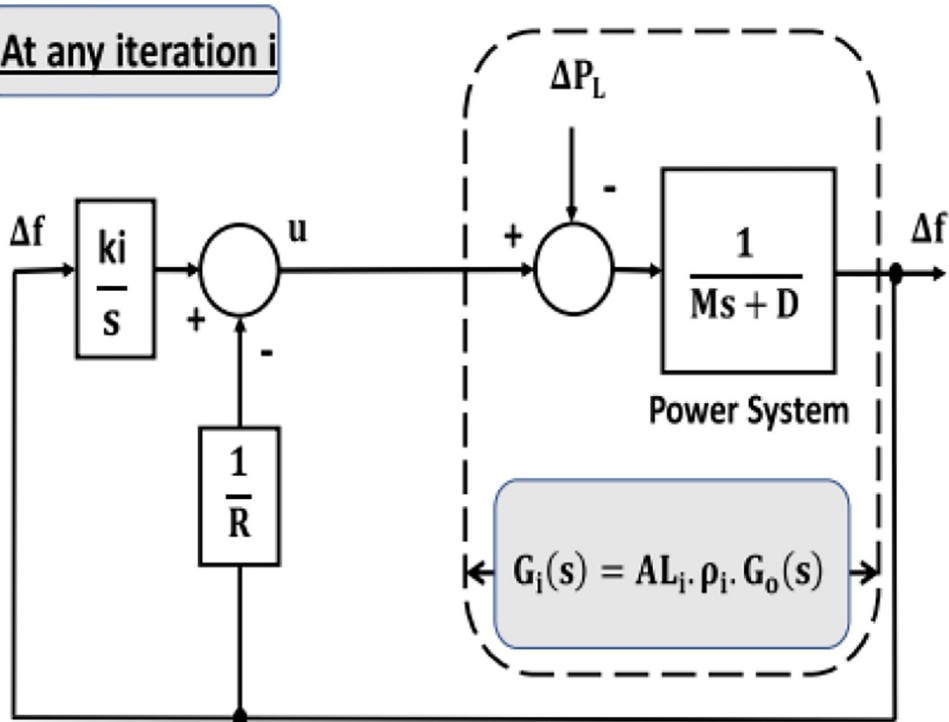

**Fig 5. Reduced model of the studied μG with the BA+BE.**

diesel power change, $\Delta f$: Frequency change, $\Delta P_L$: Load change, $\Delta P_c$: Supplementary control action, $M$: Equivalent inertia constant, $D$: Equivalent damping coefficient, $R$: Speed drop characteristic, $T_g$: Governor time constant, $T_d$: Turbine time constant, and ($\Delta f$, $\Delta P_d$, $\Delta P_g$) equal to ($\frac{df}{dt}, \frac{dPd}{dt}, \frac{dPg}{dt}$), respectively.

A second-order closed-loop system's parameters are calculated using the simplified μG model shown in Fig 5 for the controlled area to show the functionality of the proposed BA-based BE identifier.

$$\text{T.F} = \frac{wn^2}{S^2 + 2\eta Wn + Wn^2} = \frac{\frac{Ki}{Mo}}{S^2 + \left(\frac{\left(Do + \frac{1}{Ro}\right)}{Mo}\right)S + \frac{Ki}{Mo}} \tag{14}$$

where $D_o$, $R_o$, and $M_o$ are the nominal values of $D$, $R$, and $M$, respectively

$$\omega_n = \sqrt{Ki/Mo}, \quad \eta = \frac{\frac{\left(Do + \frac{1}{Ro}\right)}{Mo}}{2\omega_n} \tag{15}$$

$$T_r = \frac{\pi - \sqrt{(1-\eta^2)}}{\omega_n\sqrt{(1-\eta^2)}}, \quad T_s = \frac{4}{\eta\omega_n}, \quad M_P = e^{\frac{-\pi\eta}{\sqrt{(1-\eta^2)}}} \tag{16}$$

This is the Balloon Effect identifier's objective function based on BA.

$$J = min \sum (T_r + T_s + M_p) \tag{17}$$

To solve the system problems, the cost function $J$ is a function of $AL_i$ and $k_i$.

## 4. Stability and robustness of analysis of the suggested BA+BE scheme

The system with the proposed controller is seen as block diagram in Fig 6 and may be stated as follows:

$$\left.\begin{aligned} \dot{X}_p &= A_p X_p + b_p c_0^* r \\ y_p &= C_p^T X_p \end{aligned}\right\} \tag{18}$$

where $c_0^*$ indicates the gain of the feedforward controller at its nominal value. The regress or w can be written as follows [41, 45]:

$$w = w_m + Qe + q_n^T \tag{19}$$

$Qe = \begin{bmatrix} 0 \\ y_p^* \end{bmatrix}$ represents the error matrix, n is the output noise

$$q_n^T = [0 \quad 1] \in (\mathfrak{R}, \mathfrak{R}^{n-1}, \mathfrak{R}, \mathfrak{R}^{n+1}) = \mathfrak{R}^{2n}.$$

The suggested adaptive controller was built on a plant with a nominal transfer function; therefore, the system output will be specified as:

$$y^* = G_0(S).u \tag{20}$$

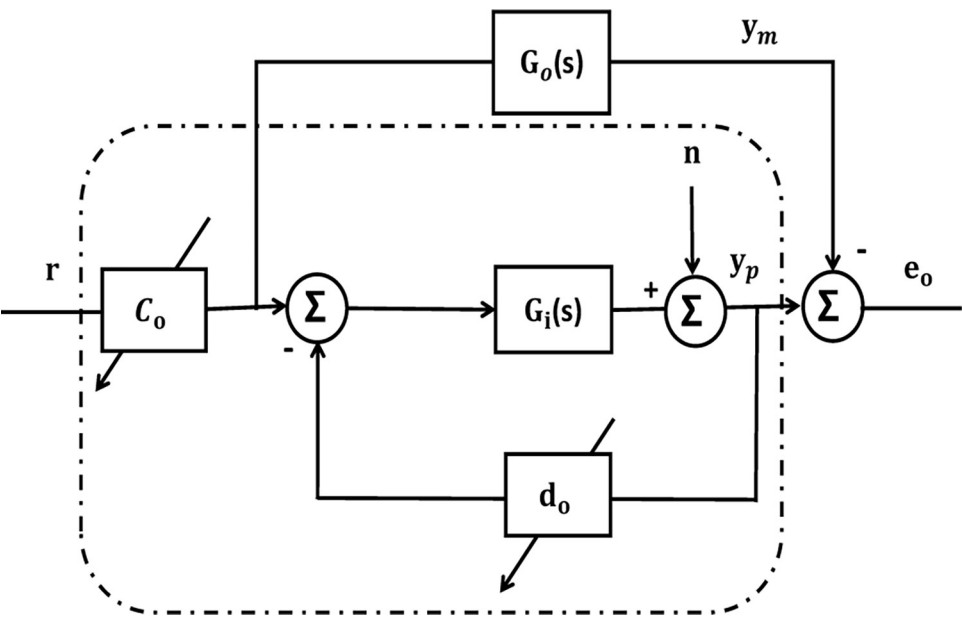

**Fig 6. Block schematic of the system for stability [46].**

The real output will be determined as follows:

$$y(t) = y^* + H_a.u(t) \tag{21}$$

$$H_a = G_i(S) - G_0(S) \tag{22}$$

If $H_a$ is an operator that satisfies that, then:

$$\|H_a.u(t)\|_\infty \leq \gamma_a \|u(t)\|_\infty + \beta_a \tag{23}$$

where $\beta_a$ and $\gamma_a$ are regarded as two constants with modest values. and for all t $\geq$ 0, $\beta_a$ covers all potential manifestations of the output's constrained disruption.

The following theorem guarantees the stability of an adaptive system even when there are unknown parameters: It is pre-summated that the adaptive system's trajectories are continuous with regard to t. If $w_m$ is Continually Exciting. Then for $x_0$, $\gamma_a$, $\beta_a$ are modest values, and the adaptive system's state trajectories are constrained.

Proof

Let T > 0 such that $x(t) \leq h$ for $t \in [0,1]$, and considering $n = H_a. u$ with assumption of:

$$\|n(t)\|_\infty \leq \gamma_a \|u(t)\|_\infty + \beta_a \tag{24}$$

For $t \in [0,T]$, and by Eq (17)

$$u = \theta^T.w = (\theta^{*T} + \emptyset^T).w = \theta^{*T}w_m + \theta^{*T}Qe + \theta^{*T}q_nn + \emptyset^T w_m + \emptyset^T Qe + \emptyset^T q_nn \tag{25}$$

and $\theta^T = [c_0 \quad d_0]$.

where $x \in B_h$, there are $\gamma_u, \beta_u \geq 0$ so,

$$\|u(t)\|_\infty \leq \gamma_u \|n(t)\|_\infty + \beta_u \tag{26}$$

For $t \in [0,T]$, and considering $\gamma_a, \beta_a$ are small values, so

$$\gamma_a\gamma_u < 1 \text{ and } \frac{\beta_a + \gamma_a\beta_u}{1 - \gamma_a\gamma_u} < c_n \tag{27}$$

where $c_n$ is assumed as a constant.

where $\beta_a$, $\gamma_u, \gamma_a$ and $\beta_u$ are not depending on T, $|x(t)| < h$ with regard to the duration. As well, a value of T > 0, can be exist as $|x(t)| \leq h$ for $t \in [0,T]$ and $x(T) = h$, moreover, this will be used for $x(T) < h$.

Using the Sphere and Matyas test functions, Fig 7(A) and 7(B) respectively, shows the ideal value of the OF ($J_{min}$) versus the number of J-evaluations. Notably, the BA method converges somewhat more quickly than the Jaya algorithm.

The limitation of the proposed method appears in the uncertainties of the governor and turbine which neglected in the simplified model that used in the design model of the proposed control method.

## 5. Results and discussions

For tuning the LFC controller of a small isolated power system, the suggested (BA+BE) approach is applied. For the simulation tests in the first section, the MATLAB/Simulink environment is employed, whereas the RT simulator is used in the second section. In Fig 8, a 20 MW diesel generator for the planned μG is depicted. In addition, Fig 9 illustrates a schematic of the proposed model with power flow. The following Tables 1 and 2, respectively list the system nominal parameters and BA parameters.

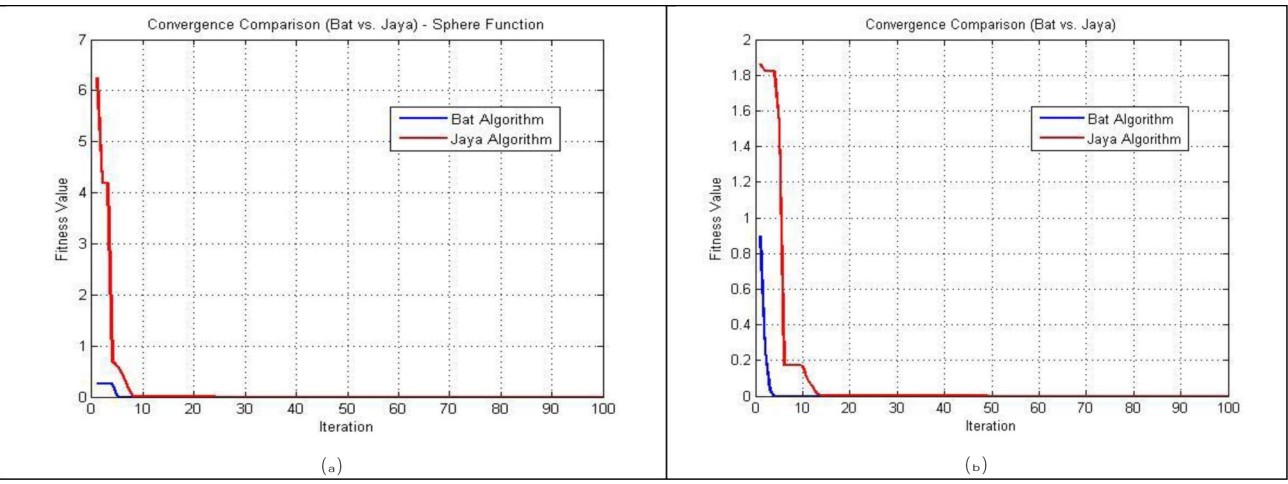

**Fig 7.** The three algorithms' convergent properties for: (a) Sphere test function; (b) Matyas test function.

## 5.1. MATLAB simulation results

**First case.** The system has been tested in this case with nominal system characteristics and a step load variation. At t = 3 seconds, the load in this instance changes from 0 pu to 0.02 pu. Both the turbine GRC and the governor dead band are taken into account. 10% per minute for the turbine GRC and 0.05 pu for the governor dead band [47]. This control strategy is assessed by contrast with I controller and Jaya+BE as reported in [45]. Tables 2, 3 contains a list of the stated parameters for the employed BA technique and Jaya algorithm, respectively. The employed Jaya optimization's stated parameters are listed in Table 3. Fig 10 shows the change in system FD for an I controller with fixed parameters, an adaptive one using the Jaya+BE optimization technique as described in [48], and an adaptive I controller employing BA+BE. Also, Fig 11 illustrates the mechanical power for this case of study. As shown in Figs 10 and 11. It can be shown that by using the proposed BA+BE, $M_p$ has been reduced by around 20%

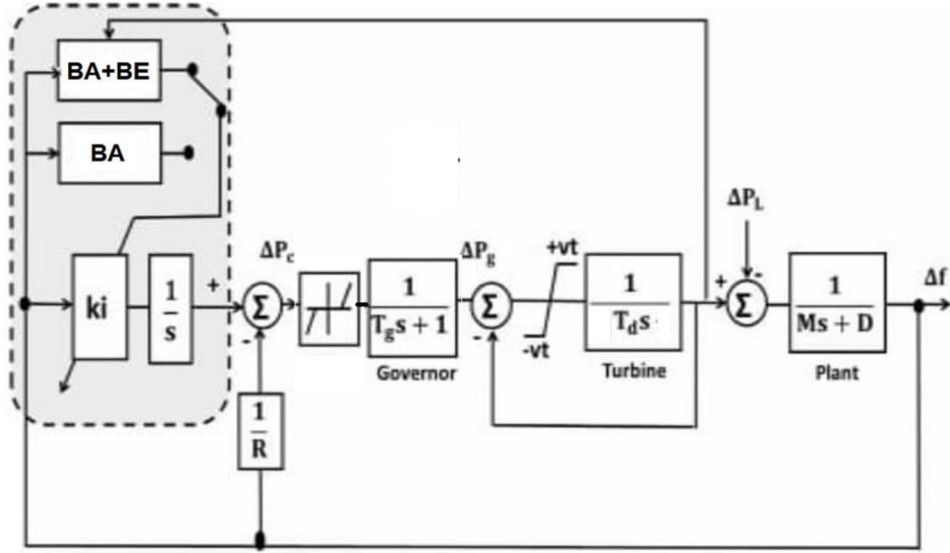

**Fig 8. Block diagram of the model of the μG power system including the proposed BA+BE.**

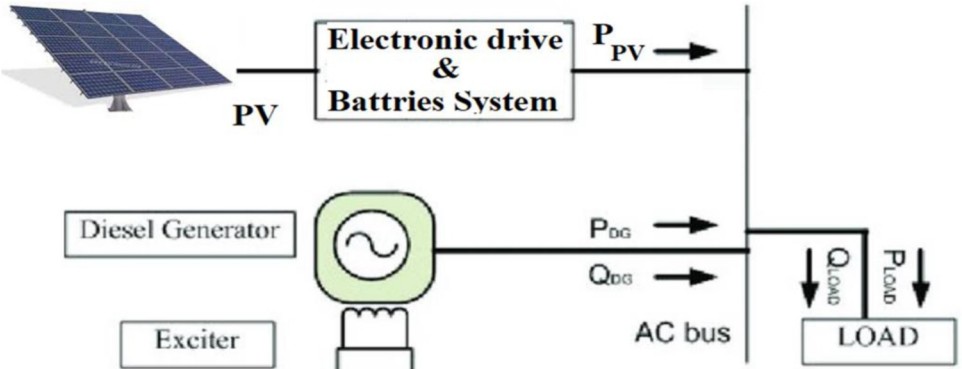

**Fig 9. A schematic of the proposed model with power flow.**

compared to the value achieved when utilizing the Jaya+BE method and I. in addition Fig 12 illustrates the values of the gains of three controllers during the simulation period. In addition, Fig 12 illustrates the values of the gains of three controllers during the simulation period.

**Second case.** In this case, a system with a proposed control scheme was examined at the case of system parameters uncertainties ($T_g$, $T_d$, D are increased by a factor of 200%). Fig 13 shows a comparison of the system response obtained with the three different controllers (fixed I, tuned by Jaya + BE, tuned by BA + BE). According to this Figure, the frequency response of the system with I controller failed to deal with this problem while systems with I adapted using Jaya + BE, BA + BE can face this case effectively. In addition, system with proposed BA + BE gives the best performance.

**Third case.** The system has been tested under various operating circumstances. Testing was done with a variable load, taking into account the system's nominal specifications and adding a 6 MW PV system to the μG as an extra source of power (with the model shown in Fig 14). According to Fig 15, the suggested (BA+BE) has the greatest performance in terms of

**Table 1. Parameters of the studied μG.**

| D | H = (M/2) | R | Tg | Td |
|---|---|---|---|---|
| (pu/Hz) | (pu. S) | (Hz/pu) | (S) | (S) |
| 0.015 | 0.08335 | 3 | 0.08 | 0.4 |

**Table 2. Suggested parameters of the BA.**

| | |
|---|---|
| Population Size (K) | 5 |
| Maximum Iteration (IT max) | 50 |
| $Q_{min}$, $Q_{max}$ | 0.5, 1 |
| The initial values of the design variables (Ki) | [.040,.025,.017,.08,.030] |

**Table 3. Data of Jaya algorithm.**

| | |
|---|---|
| Population Size (k) | 5 |
| Number of Generations (*i*) | 50 |
| Number of Design Variables (m) | 2 |

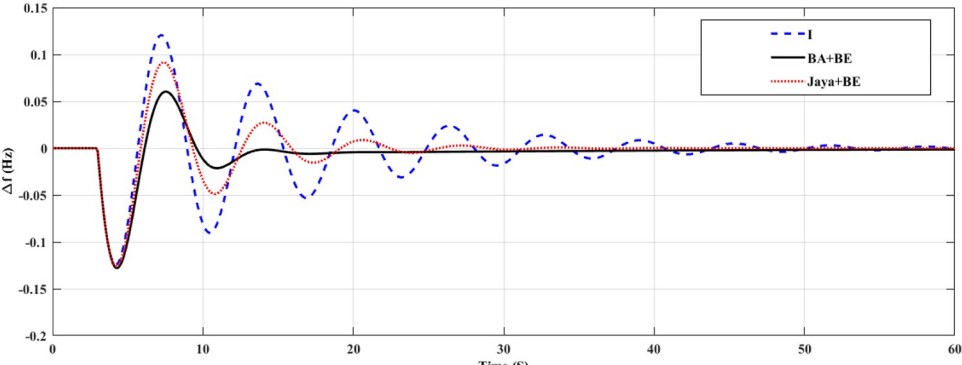

**Fig 10. System response for case 1, frequency deviation.**

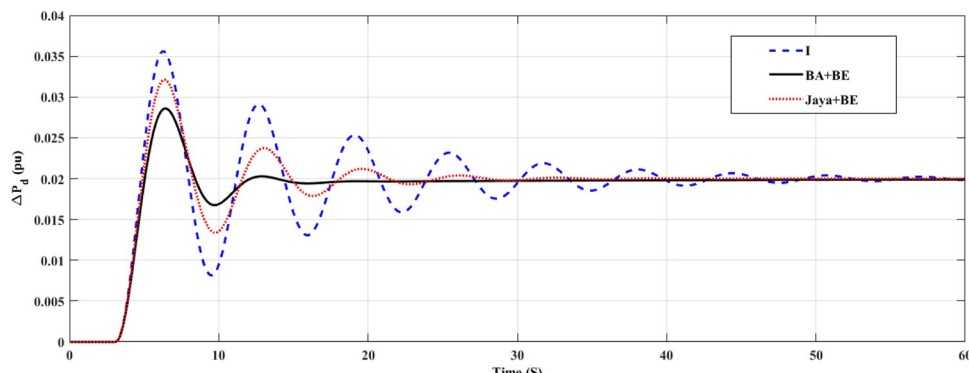

**Fig 11. System response for case 1, mechanical power.**

frequency response. These data demonstrate the advantage of the proposed online tuned (BA +BE) controller over the conventional I and Jaya+BE approach.

## 5.2. RTsimulator results

Case 1 simulation results were rerun using RTsimulator to assess the PS with the suggested controller. As illustrated in Fig 16, the proposed controller for the investigated μG-PS is

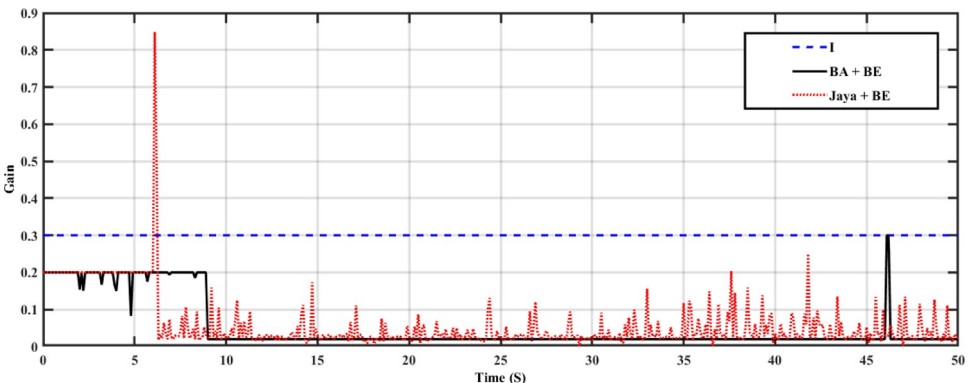

**Fig 12. The gains of three controllers.**

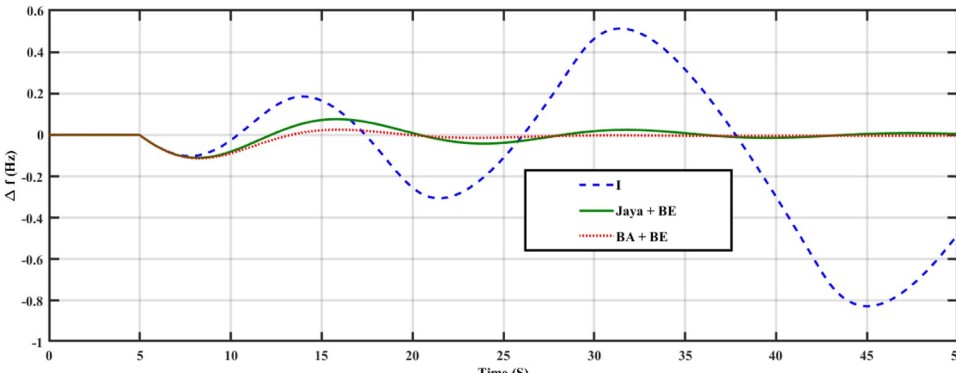

**Fig 13. System response for case 2, frequency deviation.**

incorporated on a PC that has a QUARC pid_e data acquisition card. Using a DSO-X 2014A storage oscilloscope, the output frequency signals have been captured. Fig 17 depicts the physical configuration of the system under study in a RT simulation environment. Fig 18 depicts the system's output frequency signals under identical circumstances as case 1 from before. Additionally, Fig 19 depicts the system output frequency signals in the same circumstance as in case 2 from before. All RT simulation demonstrate the usefulness of the suggested controller in comparison to the other researched controllers.

The system response achieved using three distinct controllers (fixed I, tuned by Jaya+BE Algorithm, and tuned by BA+BE) is compared in Fig 18. In the third scenario, the system has been tested under various operational circumstances as shown in Fig 19. According to Figs 18 and 19, the suggested adaptive controller using (BA+BE) has the greatest performance in terms of frequency response for the studied PS. These Figs. demonstrate the advantages of the proposed online tuned controller using (BA+BE) controller over conventional I-controllers, and adaptive one tuned using Jaya+BE methods.

## 6. Conclusions

In this study, an adaptive FR technique based on (BA+BE) is suggested to better use RESs in μG's FR operation. The BA+BE control strategy is offered for FR in circumstances of varied

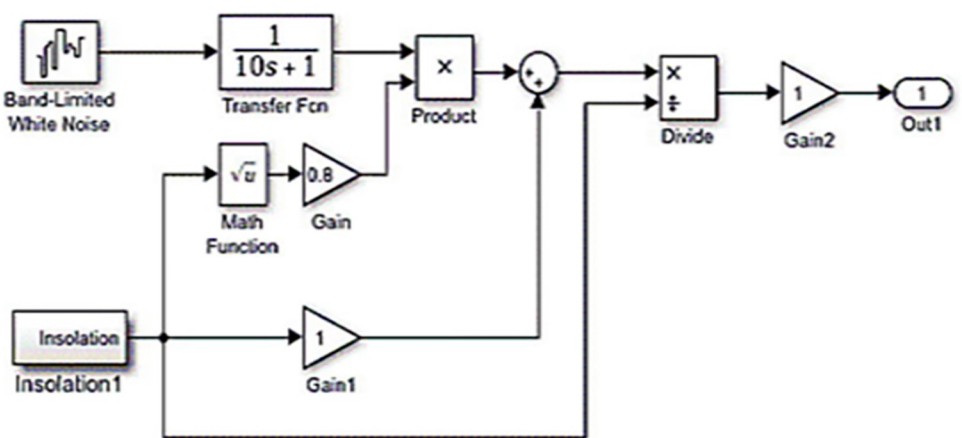

**Fig 14. Model of a variable solar PS [44].**

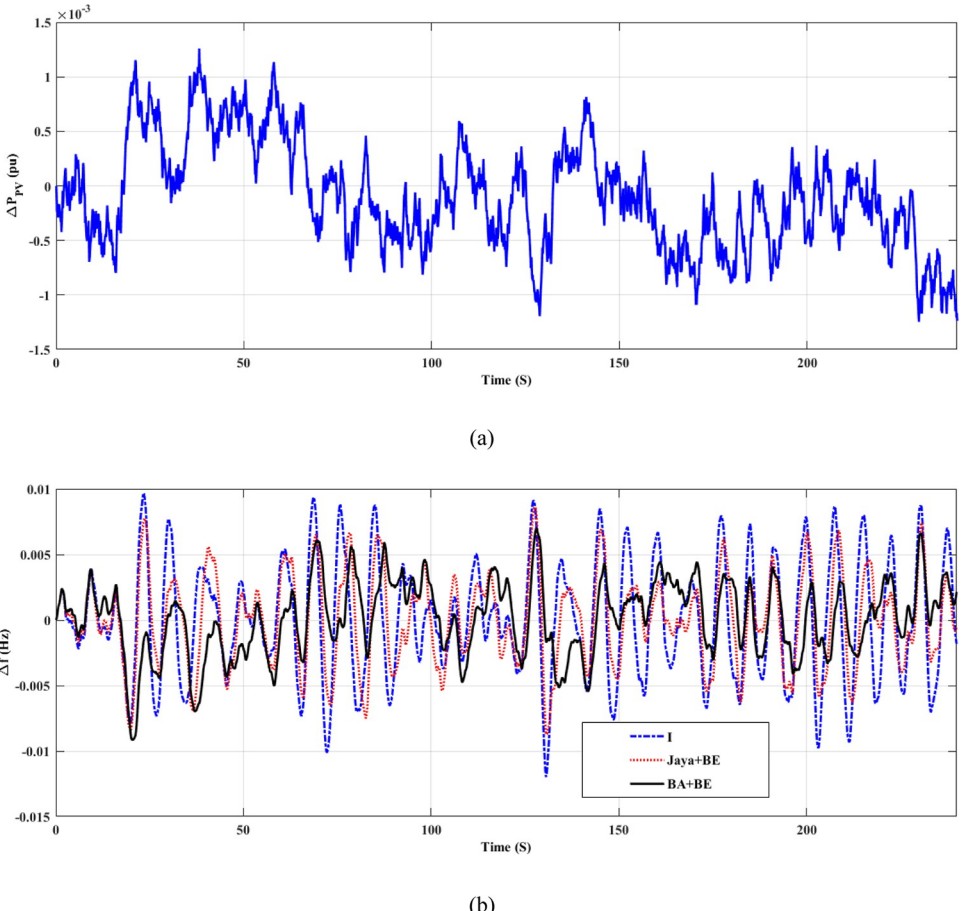

**Fig 15.** (a) PV power deviation (b) System response for case 3, FD.

disturbances and parameter uncertainties, in contrast to conventional and intelligent controllers, which are not guaranteed to work satisfactorily under a wide variety of operating conditions. The proposed technique effectively maintains the frequency constant over I and Jaya +BE by 61.5% and 31.25%, respectively. A computer simulation has been carried out to examine the suggested control mechanism under the influence of unpredictable demand loads and fluctuating PV power generation. The achieved results have been contrasted with the use of the suggested controller (BA+BE), the fixed integrated controller, and Jaya+BE on the μG, and the end results have been extensively studied. In summary, the suggested adaptive control method employing BA+BE can effectively manage significant system issues (such as disturbances and parameter fluctuations). In order to address LFC issues and minimize system oscillations, it is recommended to utilize a controller with gains that are tuned using the proposed BA+BE algorithm. Finally, a laboratory implementation of the suggested adaptive controller for variable load and variation of parameters on islanded μG has been presented. Following is a possible outline of the direction for future research:

1. Assessing the system's response to the integration of extra controllable loads with the proposed technique.

2. Research into more ideal controllers, such as the linear quadratic gaussian.

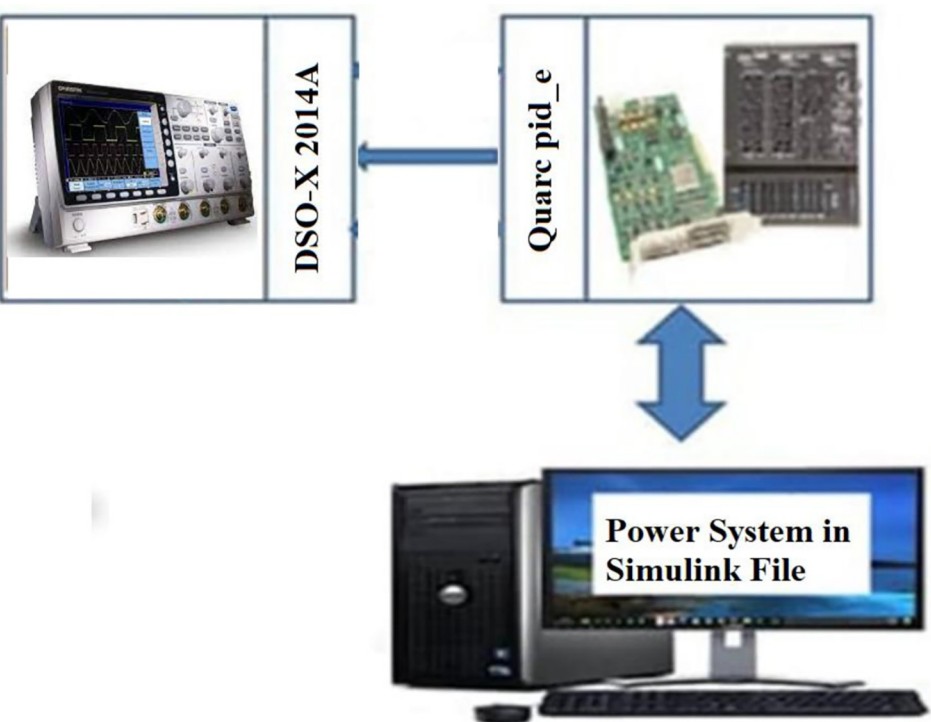

**Fig 16. Block diagram of the studied system using RT simulation.**

3. Evaluation in light of more contemporary control techniques to show the advantages and disadvantages of the investigated technique.

4. Implementation of a novel optimisation method on the system under consideration with the BE identifier.

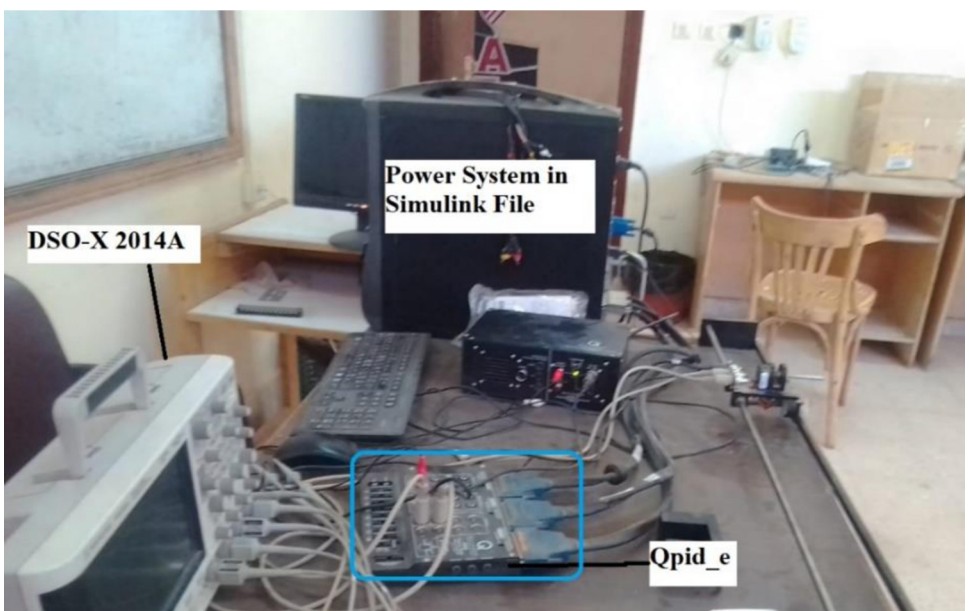

**Fig 17. RT laboratory setup.**

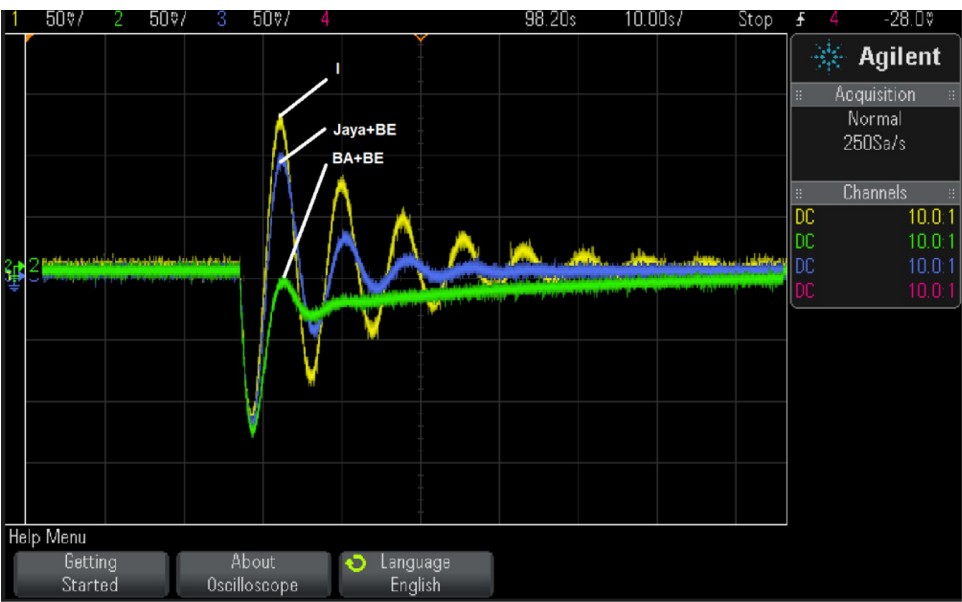

**Fig 18. System response for case 1, FD using RT simulation.**

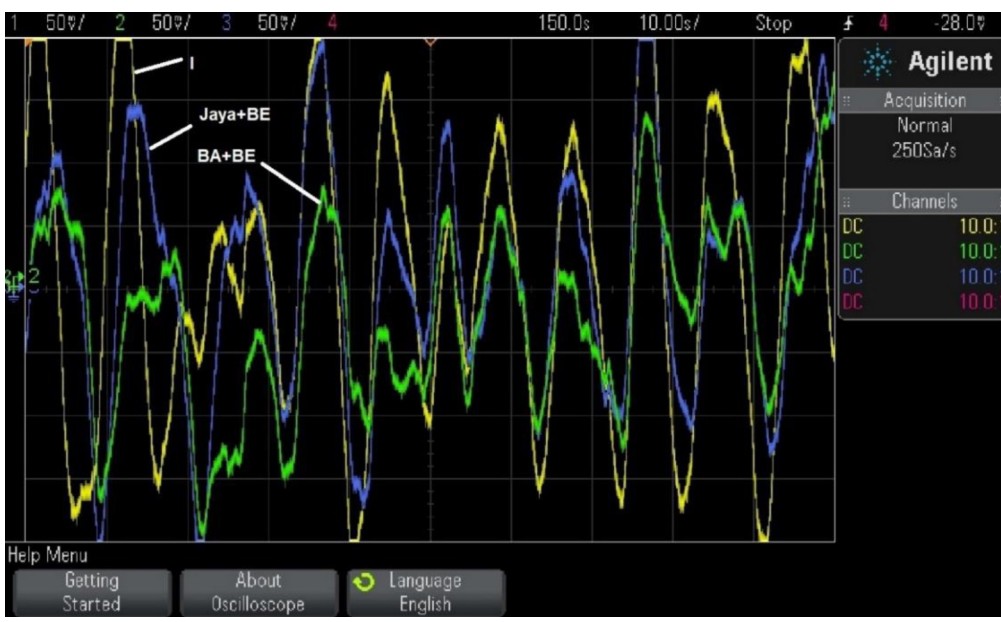

**Fig 19. System response for case 2, FD by using RT simulation.**

## Author Contributions

**Conceptualization:** Ahmed M. Ewias, Tarek Hassan Mohamed, Mohamed Metwally Mahmoud, Yasser Ahmed Dahab.

**Data curation:** Yasser Ahmed Dahab.

**Formal analysis:** Ahmed M. Ewias, Yasser Ahmed Dahab.

**Funding acquisition:** Sultan H. Hakmi, Ahmad Eid.

**Investigation:** Ahmed M. Ewias, Tarek Hassan Mohamed, Mohamed Metwally Mahmoud, Yasser Ahmed Dahab.

**Methodology:** Ahmed M. Ewias, Tarek Hassan Mohamed, Yasser Ahmed Dahab.

**Resources:** Sultan H. Hakmi, Mohamed Metwally Mahmoud, Yasser Ahmed Dahab.

**Software:** Ahmed M. Ewias, Tarek Hassan Mohamed.

**Supervision:** Ahmad Eid, Almoataz Y. Abdelaziz.

**Validation:** Mohamed Metwally Mahmoud, Yasser Ahmed Dahab.

**Writing – review & editing:** Mohamed Metwally Mahmoud, Yasser Ahmed Dahab.

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
