## [Decision Letter · Decision Letter 0]

24 Sep 2023

PONE-D-23-28961Advanced Load Frequency Control of Microgrid Using a Bat Algorithm Supported by a Balloon Effect identifier in the presence of PV power sourcePLOS ONE

Dear Dr. Mahmoud,

Thank you for submitting your manuscript to PLOS ONE. After careful consideration, we feel that it has merit but does not fully meet PLOS ONE’s publication criteria as it currently stands. Therefore, we invite you to submit a revised version of the manuscript that addresses the points raised during the review process.

We look forward to receiving your revised manuscript.

Kind regards,

Yogendra Arya

Academic Editor

PLOS ONE

Journal Requirements:

3. Thank you for stating the following financial disclosure: "No"

4. Thank you for stating the following in your Competing Interests section: "The authors declare that they have no conflict of interest concerning the publication of this article"

Please complete your Competing Interests on the online submission form to state any Competing Interests. If you have no competing interests, please state ""The authors have declared that no competing interests exist."", as detailed online in our guide for authors at http://journals.plos.org/plosone/s/submit-now.

Reviewers' comments:

Reviewer's Responses to Questions

**Comments to the Author**

1. Is the manuscript technically sound, and do the data support the conclusions?

Reviewer #1: Yes

Reviewer #2: Yes

Reviewer #3: Partly

2. Has the statistical analysis been performed appropriately and rigorously? 

Reviewer #1: Yes

Reviewer #2: Yes

Reviewer #3: No

3. Have the authors made all data underlying the findings in their manuscript fully available?

Reviewer #1: No

Reviewer #2: Yes

Reviewer #3: Yes

4. Is the manuscript presented in an intelligible fashion and written in standard English?

Reviewer #1: Yes

Reviewer #2: Yes

Reviewer #3: No

5. Review Comments to the Author

Reviewer #1: This reviewer suggests the following points to improve the paper quality:

1. Please limit acronyms in Abstract and Conclusion section (Unless the term is repeated within the section). Also avoid acronyms in Title as well as Keywords.

2. Try to include the nomenclature of all the symbols used in the work, at the beginning for better readability. Include the values of design parameters in Appendix

3. Try to redraft the Introduction section, with background, challenges, literature review, scopes, motivation, contributions, and organization of paper. Highlight the novelties/major contribution of the work prior to organization pf paper in brief (preferably in 3-bulleted points). Also try to expand the literature review including some recent works (of last 3-years) in the similar field, such as, d.o.i.: 10.1007/978-3-642-21578-0_3, 10.1016/j.energy.2021.121014, 10.1080/03772063.2022.2083026, and so on.

4. Try to maintain the work flow of the paper, especially during transition between sections and subsections.

5. Try to emphasize more on the problem statement and objective of the work.

6. Try to quote all the equations in related texts with proper citation (if adopted from published work)

7. Rectify the flowchart of BA in Fig.2, which is logically incorrect.

8. Why the author proposed BA+BE technique for this work? Justify your answer comparing the response with other contemporary methods.

9. Why only PI controllers are used to tune in this work? Justify with comparative analysis including modern controllers.

10. Please improve the resolution of all images and include xy-grids in all result plots.

11. The system responses should be analyzed including some case studies (e.g. with variation of ambient conditions) to confirm the system stability.

12. Redraft the Conclusion with numerical evidences to support your claim. Also include at least one future scope to it.

13. Try to redraft the References section with unified formatting as per the journal guidelines.

14. Proofread the entire manuscript to rectify some existing typos and grammatical errors.

Reviewer #2: The work is appreciable and publishable. The paper should be revised by incorporating the following comments:

1. Abstract should be improved. It should also have numerical data.

2. Introduction should be divided the into subsections like: 1) Background, 2) Literature review, 3) Research gap and motivation, 4) Challenges, 5) Contribution and 6) paper organization.

3. Figure are blurred. Their quality should be of high standard.

4. Fig. 6, turbine with GRC i.e., closed loop GRC model is incorrect.

5. Integral (I) is an outdated controller. In place “I”, a controller having new configuration should be used.

6. Why governor and turbine are missing in Fig 5?

7. Optimal ki gain values of the controller for each case must be given due to I, BA+BE and Jaya+BE.

8. What are the needs and benefits of this paper? What are the challenges and contributions to this field? Why a new method is needed? What are the limitations of existing ones? After a comprehensive discussion about the contributions and novelties, please, list them briefly. Please explain more clearly.

9. References to the Equations, Figures and Tables should be given.

10. Revise and refine conclusion section. Also give future directions of the work.

11. Add computation burden analysis.

12. Add convergence characteristics of decline in values of cost functions.

13. Add stability test in time/frequency domain.

14. Literature review should be strengthened by adding few more papers having DOIs like https://doi.org/10.1016/j.ijepes.2018.02.012;

https://doi.org/10.1002/2050-7038.12883;

https://doi.org/10.1016/j.seta.2022.102671;

https://doi.org/10.1049/rpg2.12061

Reviewer #3: In this study, an adaptive LFC method for power systems has been suggested to mitigate the unpredictability of the majority of green energy sources. For the tuning task, an artificial bat algorithm (BA) is utilized and is supported by the balloon effect (BE) identifier. The standard integral controller and Jaya+BE, two more optimization techniques, have been compared with the suggested BA+BE strategy. As per the results and by claimed, the suggested technique BA+BE has a significant advantage over other techniques in terms of maintaining frequency stability in the presence of step/random disturbances and PV source. The study of the paper is needing an improvement. It needs a major change in the revised manuscript.

(1) The abstract does not provide a clear problem statement subjected to the current study that the proposed algorithm is trying to solve. The abstract should highlight the significance of the problem and how the proposed algorithm addresses it.

(2) The abstract lacks details about the proposed algorithm. The abstract should provide some technical details on how the proposed algorithm works and what are the key features that make it different from the existing methods.

(3) The abstract could benefit from a more concise and straightforward writing style. The abstract should aim to convey the main ideas and findings of the paper in a clear and concise manner.

(4) In the introduction, the motivation of the paper needs to be explained for more clearly. If possible, give separate sub-section.

(5) The quality of each figure needs to be improved.

(6) It is requested to add a mathematical problem formulation.

(7) Why only an integral controller is taken into study. If possible, use PID controller for better results.

(8) What is the range of controller gains has been taken into the optimization task. It should be mentioned and how these parameters are determined, need to explain.

(9) Why the authors have chosen BA+BE particularly in this work, the details are missing whether this solution is an optimum (local or global). The simple comparisons shown in the manuscript are believed to be not sufficient.

(10) In this work, the test case should be extended to two-area power system and show the LFC response profile.

(11) The authors need to go through the entire manuscript to double-check the accuracy/consistency of each equation, table, figure, and reference and to ensure that English grammar errors are avoided.

(12) Furthermore, where are the limitations of your study? Clarifying the study's limitations allows the readers to better understand under which conditions the results should be interpreted.

(13) In terms of conclusion/implications, did your study suggest a need for further research, what might this consist of, and how might such research extend or improve the current state of knowledge in this field? Are there any practical implications that need to be addressed? All these were not highlighted in the concluding remarks.

(14) The discussion of the results needs to include the strengths and weaknesses of the proposed algorithm.

6. PLOS authors have the option to publish the peer review history of their article (what does this mean?). If published, this will include your full peer review and any attached files.

Reviewer #1: No

Reviewer #2: No

Reviewer #3: **Yes: **Chandan Kumar Shiva

---

## [Author Response · Author response to Decision Letter 0]

3 Oct 2023

***Technical response to the reviewers*** October 3th, 2023

Journal name: PLOS ONE

Manuscript No.: PONE-D-23-28961

Title: “Advanced Load Frequency Control of Microgrid Using a Bat Algorithm Supported by a Balloon Effect identifier in the presence of Photovoltaic power source”

Ahmed M. Ewias1, Sultan H. Hakmi2, Tarek Hassan Mohamed1, Mohamed Metwally Mahmoud1*, Ahmad Eid3,4, Almoataz Y. Abdelaziz5, Yasser Ahmed Dahab6

1Department of Electrical Engineering, Faculty of Energy Engineering, Aswan University, Aswan 81528, Egypt.

2Electrical Engineering Department, Faculty of Engineering, Jazan University, Jazan 45142, Saudi Arabia

3Department of Electrical Engineering, Faculty of Engineering, Aswan University, 81542 Aswan, Egypt.

4Department of Electrical Engineering, College of Engineering, Qassim University, 56452 Unaizah, Saudi Arabia.

5Faculty of Engineering and Technology, Future University in Egypt, Cairo 11835, Egypt

6Arab Academy for Science, Technology and Maritime Transport, South Valley Branch, Aswan 81516, Egypt.

ewaisa@aswu.edu.rg, shhakmi@jazanu.edu.sa, tarekhie@yahoo.com, Metwally_m@aswu.edu.eg, ahmadeid@gmail.com, almoataz.abdelaziz@fue.edu.eg, and ydahab@aast.edu

Corresponding author: Mohamed Metwally Mahmoud

Dear Editors and Reviewers

The authors are thankful to the learned Editor and Reviewers for their thoughtful and detailed comments to improve the quality of the manuscript. The authors have given reviewer comments a lot of interest in the revision process in an attempt to address all of the reviewers’ concerns and corrections as you will already find them incorporated in the revised manuscript. Moreover, a reply to each of the reviewers’ comments is provided below.

Kindly find the response to the reviewer’s comments in the following paragraphs. We hope this revised version of the manuscript meets the editor and reviewers’ expectations, and the standards of publication in the PLOS ONE Journal. 

The changes carried out by the authors are incorporated in the revised manuscript and highlighted in YELLOW.

Editor's Comments:

Comments to the Authors:

Comment-1: Thank you for submitting your manuscript to PLOS ONE. After careful consideration, we feel that it has merit but does not fully meet PLOS ONE’s publication criteria as it currently stands. Therefore, we invite you to submit a revised version of the manuscript that addresses the points raised during the review process. Please submit your revised manuscript by Nov 08 2023 11:59PM.

Response-1: Our sincere thanks and appreciation to the editor for considering our manuscript for publication in PLOS ONE Journal, and the recommending submission of the revised manuscript. To improve the quality of the manuscript, the reviewer's queries are addressed and their suggestions are incorporated into the revised manuscript. A new method called hybrid bat algorithm-balloon effect identifier optimizer is designed and implemented and compared with integral controller and Jaya+BE for frequency stability. The introduction section is rewritten, many changes in title, abstract, simulation results, and conclusions are done based on reviewers’ quires. Keywords are rearranged in aphetic order. Some sentences have been edited in the revised paper to clarify the paper's contributions and enhance the paper quality.

Comment-2: Please include the following items when submitting your revised manuscript:

Response-2: Our sincere thanks and appreciation to the editor for his comment. The required items are attached during submission process. A cover letter is provided and prepared to explain, point by point, the details of the revisions to the manuscript. The changes carried out by the authors are incorporated in the revised manuscript and highlighted in YELLOW to be easily viewed by the editors and reviewers. An unmarked version of the revised paper without tracked changes is also provided.

Comment-3: Please ensure that your manuscript meets PLOS ONE's style requirements.

Response-3: The authors are extremely thankful to the editor for this thoughtful point. The revised manuscript meets PLOS ONE's style.

Reviewers Comments:

Reviewer 1 

Comments to the Authors:

Comment-1: Please limit acronyms in Abstract and Conclusion section (Unless the term is repeated within the section). Also avoid acronyms in Title as well as Keywords.

Response-1: At the beginning, the authors are thankful to the honorable reviewer for the words of encouragement and trust in our work. The acronyms are limited in abstract and conclusion in the updated version. In addition, acronyms in title and keywords are avoided. Kindly check the revised manuscript.

Comment-2: Try to include the nomenclature of all the symbols used in the work, at the beginning for better readability. Include the values of design parameters in Appendix

Response-2: The authors are thankful to the esteemed reviewer upon his valuable comment. Based on the reviewer comment, the authors include the nomenclature of all the symbols used in this work, at the beginning for better readability. The values of design parameters are provided in the updated paper. Kindly check the revised manuscript.

Comment-3: Try to redraft the Introduction section, with background, challenges, literature review, scopes, motivation, contributions, and organization of paper. Highlight the novelties/major contribution of the work prior to organization pf paper in brief (preferably in 3-bulleted points). Also try to expand the literature review including some recent works (of last 3-years) in the similar field, such as, d.o.i.: 10.1007/978-3-642-21578-0_3, 10.1016/j.energy.2021.121014, 10.1080/03772063.2022.2083026, and so on.

Response-3: The authors are thankful to the esteemed reviewer upon his valuable comment. The suggested references enhance the introduction section specially the literature review part. All the suggested references are added in the updated paper [1], [2], [3]. Kindly check the revised manuscript.

[1] R. Majumder, A. Ghosh, and G. Ledwich, “Load Frequency Control in a Microgrid: Challenges and Improvements,” Power Syst., vol. 53, pp. 49–82, 2012, doi: 10.1007/978-3-642-21578-0_3.

[2] Q. Yang, N. Dong, and J. Zhang, “An enhanced adaptive bat algorithm for microgrid energy scheduling,” Energy, vol. 232, 2021, doi: 10.1016/j.energy.2021.121014.

[3] M. Bhuyan, D. C. Das, A. K. Barik, and S. C. Sahoo, “Performance Assessment of Novel Solar Thermal-Based Dual Hybrid Microgrid System Using CBOA Optimized Cascaded PI-TID Controller,” IETE J. Res., 2022, doi: 10.1080/03772063.2022.2083026.

Comment-4: Try to maintain the work flow of the paper, especially during transition between sections and subsections.

Response-4: The authors are thankful to the esteemed reviewer upon his valuable comment. The work flow of the revised paper is maintained during transition between sections and subsections. Kindly check the revised manuscript.

Comment-5: Try to emphasize more on the problem statement and objective of the work.

Response-5: The authors are thankful to the esteemed reviewer upon his valuable comment. The problem statement and objective of the work is emphasized in the updated paper. Kindly check the revised manuscript.

Comment-6: Try to quote all the equations in related texts with proper citation (if adopted from published work)

Response-6: The authors are thankful to the esteemed reviewer upon his valuable comment. All the equations are quoted in related texts in the updated paper. Kindly check the revised manuscript.

Comment-7: Rectify the flowchart of BA in Fig.2, which is logically incorrect.

Response-7: The authors are thankful to the esteemed reviewer upon his valuable comment. The flowchart of BA has been replaced by corrected one. The suggested flowchart of BA in Fig.2 is rectified in the updated paper. Kindly check the revised manuscript.

Comment-8: Why the author proposed BA+BE technique for this work? Justify your answer comparing the response with other contemporary methods.

Response-8: The authors are thankful to the esteemed reviewer upon his valuable comment. Many industrial applications have employed the bat optimization algorithm (BA) to adaptively adjust the gains of traditional controllers [23]. A cascaded PI-fractional order PID (PI-FOPID) controller with fine-tuned BA can improve the hybrid μG system frequency response [24]. In order to solve this issue and increase the optimization algorithm's sensitivity to both disturbances and parameter changes, a balloon effect (BE) adjustment proved its effectiveness as provided in [25] so, it is suggested for this study. Actually, the proposed design of BA+BE is single input/single output system so it is suitable for isolated micro grid. A section of convergence has been added to support the priority of BA comparing with Jaya which used before for LFC issue. Kindly check the revised manuscript.

Comment-9: Why only PI controllers are used to tune in this work? Justify with comparative analysis including modern controllers.

Response-9: The authors are thankful to the esteemed reviewer upon his valuable comment. According to trustworthy references; to make a design of classical controller of LFC of single area power system, Integrator controller is the suitable one for the closed loop transfer function of the proposed system. Secondly it is suitable for the physical nature of the gate of the governor. Because of the physical nature of the governor input (where the governor does not responses to fast changes in its input signals) so Integral controller is the best conventional controller suitable for LFC application. 

Comment-10: Please improve the resolution of all images and include xy-grids in all result plots.

Response-10: The authors are thankful to the esteemed reviewer upon his valuable comment. The resolution of all images in the updated paper is improved. Kindly check the revised manuscript.

Comment-11: The system responses should be analysed including some case studies (e.g. with variation of ambient conditions) to confirm the system stability.

Response-11: The authors are thankful to the esteemed reviewer upon his valuable comment. A case of system parameters changes has been added in the updated paper. This scenario proves the accuracy of the proposed technique. Kindly check the revised manuscript.

Comment-12: Redraft the Conclusion with numerical evidences to support your claim. Also include at least one future scope to it.

Response-12: The authors are thankful to the esteemed reviewer upon his valuable comment. Redrafting the conclusion with numerical evidences as well as future scope is done in the revised paper. Kindly check the revised manuscript.

Comment-13: Try to redraft the References section with unified formatting as per the journal guidelines.

Response-13: The authors are thankful to the esteemed reviewer upon his valuable comment. Redrafting the references section with unified formatting as per the journal guidelines is done in the revised paper. Kindly check the revised manuscript.

Comment-14: Proofread the entire manuscript to rectify some existing typos and grammatical errors.

Response-14: The authors are thankful to the esteemed reviewer upon his valuable comment. Proofreading the revised manuscript is done. Kindly check the revised manuscript. 

Reviewer 2 

Comment-1: The work is appreciable and publishable. The paper should be revised by incorporating the following comments:

1. Abstract should be improved. It should also have numerical data.

Response-1: At the beginning, the authors are thankful to the honorable reviewer for the words of encouragement and trust in our work. The abstract is improved and have numerical data in the updated version. Kindly check the revised manuscript.

Comment-2: Introduction should be divided the into subsections like: 1) Background, 2) Literature review, 3) Research gap and motivation, 4) Challenges, 5) Contribution and 6) paper organization.

Response-2: The authors are thankful to the esteemed reviewer upon his valuable comment. All the required suggestions are done in the revised version. Kindly check the revised manuscript.

Comment-3: Figure are blurred. Their quality should be of high standard.

Response-3: The authors are extremely thankful to the reviewer for this thoughtful point. Based on the esteemed reviewer suggestion, all the figures quality are enhanced in the updated version. Kindly check the revised manuscript. 

Comment-4: Fig. 6, turbine with GRC i.e., closed loop GRC model is incorrect.

Response-4: The authors are extremely thankful to the reviewer for this thoughtful point. The Figure is corrected in the updated paper. Kindly check the revised manuscript.

Comment-5: Integral (I) is an outdated controller. In place “I”, a controller having new configuration should be used.

Response-5: The authors are extremely thankful to the reviewer for this thoughtful point. Because of the physical nature of the governor input (where the governor does not responses to fast changes in its input signals) so Integral controller is the best conventional controller suitable for LFC application

Comment-6: Why governor and turbine are missing in Fig 5?

Response-6: The authors are extremely thankful to the reviewer for this thoughtful point. Comparing to the value of inertia (M and D) the dynamic of governor and turbine can be neglected in the reduced system model to simplify the design operation. Kindly check the revised manuscript.

Comment-7: Optimal ki gain values of the controller for each case must be given due to I, BA+BE and Jaya+BE.

Response-7: The authors are extremely thankful to the reviewer for this thoughtful point. As suggested by the esteemed reviewer, Fig. 12 for the gains of the controllers has been added in the updated paper. Kindly check the revised manuscript.

Comment-8: What are the needs and benefits of this paper? What are the challenges and contributions to this field? Why a new method is needed? What are the limitations of existing ones? After a comprehensive discussion about the contributions and novelties, please, list them briefly. Please explain more clearly.

Response-8: The authors are extremely thankful to the reviewer for this thoughtful point. The needs and benefits of this study is provided in the updated paper. The proposed method did not appear in any published research article, so it is needed to investigate this method performance.

Despite the good response achieved by using algorithms, the door is opened to more enhancements in system responses during difficulties such as parameters changes and penetrations resulted from renewable power sources. Addition explanation has been added to clarify the need of the proposed algorithm.

This research suggests an innovative adaptive LFC technique to enhance the degree of PV participation in μG in order to address the aforementioned difficulties. This paper proposes an adaptive LFC scheme for fluctuating loads and parameters in smart μG, based on a BA with BE (BA+BE). Diesel generators, electrical load, and PV make up the μG that is being considered. The influence of FD brought on by both random demand loads and RESs is investigated in order to evaluate the proposed (BA+BE) optimizer. In order to demonstrate its accuracy and robustness, it is also contrasted with the traditional integral (I) controller and Jaya approaches. A thorough simulation and real-time (RT) investigation are undertaken to confirm the successful application of the idea, and the suggested control strategy's detailed design process and implementation structure are described. A laboratory implementation of the desired controller with the studied system is presented. In this step, the BA+BE, I, and Jaya+BE algorithms are applied to RT implementation using a PC with QUARC pid_e data acquisition card and MATLAB software with QUARC sub-program. Using a storage oscilloscope, the system frequency and algorithm outputs are recorded. The main outstanding features of this work can be expressed as follows:

• Using the BA+BE optimizer, which is fed by the output of the open-loop simplified µG transfer function, an online adaptable LFC is investigated.

• This paper demonstrates the efficiency of a BA+BE optimizer-adjusted I controller in LFC issues.

• The performance of the suggested adaptive approach is superior to that of the traditional I and Jaya+BE controllers.

Comment-9: References to the Equations, Figures and Tables should be given.

Response-9: The authors are extremely thankful to the reviewer for this thoughtful point. As suggested by the esteemed reviewer, References to the Equations, Figures and Tables are given in the updated paper. Kindly check the revised manuscript.

Comment-10: Revise and refine conclusion section. Also give future directions of the work.

Response-10: The authors are thankful to the esteemed reviewer upon his valuable comment. Redrafting the conclusion with numerical evidences as well as future directions is done in the revised paper. Kindly check the revised manuscript.

Comment-11: Add computation burden analysis.

Response-11: The authors are thankful to the esteemed reviewer upon his valuable comment. As suggested by the esteemed reviewer, the main advantage of mixing the balloon effect identifier with the optimization method in adaptive case is to absorb large range of system parameters uncertainties and system disturbances, this leads to improve computation process and decrease its burden. Kindly check the revised manuscript.

Comment-12: Add convergence characteristics of decline in values of cost functions.

Response-12: The authors are thankful to the esteemed reviewer upon his valuable comment. As suggested by the esteemed reviewer, a section of convergence has been added in the revised paper. Kindly check the revised manuscript.

Comment-13: Add stability test in time/frequency domain.

Response-13: The authors are thankful to the esteemed reviewer upon his valuable comment. As suggested by the esteemed reviewer, a section of stability test has been added in the revised paper. Kindly check the revised manuscript.

Comment-14: Literature review should be strengthened by adding few more papers having DOIs like https://doi.org/10.1016/j.ijepes.2018.02.012;

https://doi.org/10.1002/2050-7038.12883;

https://doi.org/10.1016/j.seta.2022.102671;

https://doi.org/10.1049/rpg2.12061

Response-14: The authors are thankful to the esteemed reviewer upon his valuable comment. As suggested by the esteemed reviewer, the suggested references have been added in the updated paper. Kindly check the revised manuscript.[4], [5], [6]

[4] P. Dahiya, P. Mukhija, and A. R. Saxena, “Design of sampled data and event-triggered load frequency controller for isolated hybrid power system,” Int. J. Electr. Power Energy Syst., vol. 100, pp. 331–349, 2018, doi: 10.1016/j.ijepes.2018.02.012.

[5] R. Choudhary, J. N. Rai, and Y. Arya, “Cascade FOPI-FOPTID controller with energy storage devices for AGC performance advancement of electric power systems,” Sustain. Energy Technol. Assessments, vol. 53, 2022, doi: 10.1016/j.seta.2022.102671.

[6] Y. Arya et al., “Cascade-IλDμN controller design for AGC of thermal and hydro-thermal power systems integrated with renewable energy sources,” IET Renew. Power Gener., vol. 15, no. 3, pp. 504–520, 2021, doi: 10.1049/rpg2.12061.

Reviewer 3

Comment-1: The abstract does not provide a clear problem statement subjected to the current study that the proposed algorithm is trying to solve. The abstract should highlight the significance of the problem and how the proposed algorithm addresses it.

Response-1: The authors are thankful to the esteemed reviewer upon his valuable comment. As suggested by the esteemed reviewer, the abstract is rewritten and involve a numerical data. Kindly check the revised manuscript

Comment-2: The abstract lacks details about the proposed algorithm. The abstract should provide some technical details on how the proposed algorithm works and what are the key features that make it different from the existing methods.

Response-2: The authors are thankful to the esteemed reviewer upon his valuable comment. As suggested by the esteemed reviewer, the abstract is rewritten to provide some technical details on how the proposed technique works and involve a numerical data. Kindly check the revised manuscript

Comment-3: The abstract could benefit from a more concise and straightforward writing style. The abstract should aim to convey the main ideas and findings of the paper in a clear and concise manner.

Response-3: Thank you for this comment. As suggested by the esteemed reviewer, the updated abstract is rewritten and become more concise and clearer. Kindly check the revised manuscript

Comment-4: In the introduction, the motivation of the paper needs to be explained for more clearly. If possible, give separate sub-section.

Response-4: The authors are thankful to the esteemed reviewer upon his valuable comment. As suggested by the esteemed reviewer, the motivation of the updated paper has been explained. Kindly check the revised manuscript.

Comment-5: The quality of each figure needs to be improved.

Response-5: The authors are extremely thankful to the reviewer for this thoughtful point. The Figures quality have been improved in the updated paper. Kindly check the revised manuscript.

Comment-6: It is requested to add a mathematical problem formulation.

Response-6: The authors are extremely thankful to the reviewer for this thoughtful point. A new section of problem formulation has been added in the updated paper. Kindly check the revised manuscript.

Comment-7: Why only an integral controller is taken into study. If possible, use PID controller for better results.

Response-7: The authors are extremely thankful to the reviewer for this thoughtful point. An integral controller is taken into study because of the physical nature of the governor input (where the governor does not responses to fast changes in its input signals) so Integral controller is the best conventional controller suitable for LFC application.

Comment-8: What is the range of controller gains has been taken into the optimization task. It should be mentioned and how these parameters are determined, need to explain.

Response-8: The authors are extremely thankful to the reviewer for this thoughtful point. The range of controller gains has been added for BA+BE are 0.001 and 1 as listed in table 2, and it has been taken around the designed value of the conventional one which designed according the nominal values of the system parameters. Kindly check the revised manuscript.

Comment-9: Why the authors have chosen BA+BE particularly in this work, the details are missing whether this solution is an optimum (local or global). The simple comparisons shown in the manuscript are believed to be not sufficient.

Response-9: The authors are extremely thankful to the reviewer for this thoughtful point. A section of convergence has been added to support the priority of Bat algorithm comparing with Jaya which used before for LFC issue. Kindly check the revised manuscript.

Comment-10: In this work, the test case should be extended to two-area power system and show the LFC response profile.

Response-10: The authors are extremely thankful to the reviewer for this thoughtful point. Actually, the proposed design of BA+BE is single input/single output system so it is suitable for isolated micro grid, in fact, for future work, we are working to modify the mathematical model of BA+BE to be extended multi-input/multi output applications such as interconnected power systems. Kindly check the revised manuscript.

Comment-11: The authors need to go through the entire manuscript to double-check the accuracy/consistency of each equation, table, figure, and reference and to ensure that English grammar errors are avoided.

Response-11: The authors are thankful to the esteemed reviewer upon his valuable comment. Proofreading the revised manuscript is done. Kindly check the revised manuscript. 

Comment-12: Furthermore, where are the limitations of your study? Clarifying the study's limitations allows the readers to better understand under which conditions the results should be interpreted.

Response-12: The authors are extremely thankful to the reviewer for this thoughtful point. The limitation of the proposed method appears in the uncertainties of the governor and turbine which neglected in the simplified model that used in the design model of the proposed control method.

Comment-13: In terms of conclusion/implications, did your study suggest a need for further research, what might this consist of, and how might such research extend or improve the current state of knowledge in this field? Are there any practical implications that need to be addressed? All these were not highlighted in the concluding remarks.

Response-13: The authors are extremely thankful to the reviewer for this thoughtful point. The esteemed reviewer suggestions are incorporated in the updated paper. Kindly check the revised manuscript.

Comment-14: The discussion of the results needs to include the strengths and weaknesses of the proposed algorithm.

Response-14: The authors are extremely thankful to the reviewer for this thoughtful point. The esteemed reviewer suggestions are incorporated in the updated paper. Kindly check the revised manuscript.

The authors once again thank the learned Editors and Reviewers for their valuable comments for improving the quality of the manuscript.

---

## [Decision Letter · Decision Letter 1]

10 Oct 2023

Advanced Load Frequency Control of Microgrid Using a Bat Algorithm Supported by a Balloon Effect Identifier in the presence of Photovoltaic Power Source

PONE-D-23-28961R1

Dear Dr. Mahmoud,

We’re pleased to inform you that your manuscript has been judged scientifically suitable for publication and will be formally accepted for publication once it meets all outstanding technical requirements.

Kind regards,

Yogendra Arya

Academic Editor

PLOS ONE

Additional Editor Comments (optional):

The reviewers have favourable with the content of the paper.

Hence, paper may be considered.

Reviewers' comments:

Reviewer's Responses to Questions

**Comments to the Author**

1. If the authors have adequately addressed your comments raised in a previous round of review and you feel that this manuscript is now acceptable for publication, you may indicate that here to bypass the “Comments to the Author” section, enter your conflict of interest statement in the “Confidential to Editor” section, and submit your "Accept" recommendation.

Reviewer #1: All comments have been addressed

Reviewer #3: All comments have been addressed

2. Is the manuscript technically sound, and do the data support the conclusions?

Reviewer #1: Yes

Reviewer #3: Yes

3. Has the statistical analysis been performed appropriately and rigorously? 

Reviewer #1: N/A

Reviewer #3: Yes

4. Have the authors made all data underlying the findings in their manuscript fully available?

Reviewer #1: Yes

Reviewer #3: Yes

5. Is the manuscript presented in an intelligible fashion and written in standard English?

Reviewer #1: Yes

Reviewer #3: Yes

6. Review Comments to the Author

Reviewer #1: All concerns were addressed in the revised manuscript. However, a thorough proofreading is recommended to rectify typos. Also, unify the reference formats as per the journal guidelines.

Reviewer #3: In this work, an adaptive load frequency control (LFC) method for power systems has been studied and suggested to mitigate the frequency deviation problem. Here, bat algorithm (BA) is used to address the LFC issue. For online gain tuning, an integral controller using an artificial BA is utilized, and this control method is supported by a modification known as the balloon effect (BE) identifier. Also, Stability and robustness of analysis of the suggested BA+BE scheme is investigated. In order to validate the MATLAB simulation results, real-time simulation tests are given utilizing a PC and a QUARC pid_e data acquisition card. The presentation and work of this paper is well convincing. Some logical outcome has come out of the paper. The obtained results show the growth and understanding in the field of LFC optimization. I have the acceptance of this paper in the present form.

7. PLOS authors have the option to publish the peer review history of their article (what does this mean?). If published, this will include your full peer review and any attached files.

Reviewer #1: No

Reviewer #3: **Yes: **Chandan Kumar Shiva

---

## [Editor Report · Acceptance letter]

13 Oct 2023

PONE-D-23-28961R1 

Advanced Load Frequency Control of Microgrid Using a Bat Algorithm Supported by a Balloon Effect Identifier in the presence of Photovoltaic Power Source 

Dear Dr. Mahmoud:

I'm pleased to inform you that your manuscript has been deemed suitable for publication in PLOS ONE. Congratulations! Your manuscript is now with our production department. 

Kind regards, 

on behalf of

Dr. Yogendra Arya 

Academic Editor

PLOS ONE